

# GAN inversion and shifting: recommending product modifications to sellers for better user preference

Satyadwyoom Kumar[1], Abhijith Sharma[2] and Apurva Narayan[3]

[1] Computer Science, Netaji Subhas Institute of Technology, New Delhi, Delhi, India
[2] Electrical and Computer Engineering, University of Waterloo, Waterloo, Ontario, Canada
[3] Computer Science, University of Western Ontario, London, Ontario, Canada

## ABSTRACT

In efforts to better accommodate users, numerous researchers have endeavored to model customer behavior, seeking to comprehend how they interact with diverse items within online platforms. This exploration has given rise to recommendation systems, which utilize customer similarity with other customers or customer-item interactions to suggest new items based on the existing item catalog. Since these systems primarily focus on enhancing customer experiences, they overlook providing insights to sellers that could help refine the aesthetics of their items and increase their customer coverage. In this study, we go beyond customer recommendations to propose a novel approach: suggesting aesthetic feedback to sellers in the form of refined item images informed by customer-item interactions learned by a recommender system from multiple consumers. These images could serve as guidance for sellers to adapt existing items to meet the dynamic preferences of multiple users simultaneously. To evaluate the effectiveness of our method, we design experiments showcasing how changing the number of consumers and the class of item image used affect the change in preference score. Through these experiments, we found that our methodology outperforms previous approaches by generating distinct, realistic images with user preference higher by 16.7%, thus bridging the gap between customer-centric recommendations and seller-oriented feedback.

## INTRODUCTION

With the availability of large amounts of data, businesses have started utilizing data-driven insights to distill their products (*Rosa, Rodriguez & Bressan, 2015*; *He & McAuley, 2016*; *Kang et al., 2017*). This distillation fosters the development of carefully designed products to improve their saleability and increase their competitiveness in the market. Moreover, it also helps in enhancing a consumer's experience leading to mutually beneficial situation for both consumers and sellers. A major tool from the artificial intelligence (AI) domain supporting a portion of this operation across businesses is recommendation systems which have been extensively used to match users to a relevant catalog of items. Empirically recommendation systems unravel the hidden similarity between features extracted from historical data such as past clicked/preferred items or user-user affinity available for a

Corresponding author
Apurva Narayan,
apurva.narayan@uwo.ca

user and a set of items currently untouched by that user on the given platform. After this, it ranks this unseen set of items based on a variety of user-centric metrics to fit a user's dynamic preference.

A significant limitation of current recommendation systems is their exclusive focus on users, primarily retrieving new items from an existing catalog that aligns with user preferences. This consumer-centric approach neglects the provision of feedback to item vendors. Introducing a feedback mechanism for product vendors, such as fashion designers, real estate agents, and hospitality businesses, would enable them to gain deeper insights into their customer base. Consequently, this would allow them to make generic refinements in their product design that enhance their appeal to a wider audience, increasing the product's likelihood of commercial success.

In the seminal work by *Goodfellow et al. (2014)*, the author introduced a novel class of neural networks known as generative adversarial networks (GANs). GANs operate on the principles of game theory (zero-sum game), to produce synthetic images using actual images as references. Initially, the images generated by this approach were of low resolution and poor quality. However, subsequent advancements by researchers such as *Karras et al. (2017)*, *Brock, Donahue & Simonyan (2018)*, and *Karras et al. (2020)* addressed these limitations, achieving the generation of high-resolution, photorealistic images. These developments have enabled the AI community to leverage GANs to enhance neural network generalizability by training it on novel data points. Furthermore, GANs have been extensively explored in various applications, including image inpainting, style transfer, and the manipulation of object attributes (*e.g.*, altering facial features), underscoring their broad applicability and utility in diverse domains.

To utilize the explorative power of GANs (*Goodfellow et al., 2014*), *Kang et al. (2017)* propose fetching novel item images out of a convolutional neural network-based generator. This proposed generator model incorporates a user and class feature to generate image samples having a high preference score for this user. A user's preference score is a value that quantifies whether a user would favor or select a given generated item. Further, the images generated in *Kang et al. (2017)* inherited a lot of visual artifacts. The artifacts, as pointed out by works like *Brown et al. (2017a)*, *Liu et al. (2019)*, *Kumar & Narayan (2022)*, *Jiang et al. (2018)*, and *Jang et al. (2019)* provide a false sense of high user preference as the generator during the optimization process may get biased towards maximizing a user's preference score.

Though *Kang et al. (2017)* propose to explore new horizons within item recommendation from a user perspective, their setup does not take into account scenarios where the generated item could belong to multiple classes at a time, eg: sandals + shoes to generate an entirely novel product that crosses the boundaries of human imagination. Additionally considering a single class while generating images will have heavy implications from a seller's standpoint as any newly generated image would not be appealing to a person who is looking for a different class item, reducing the item's consumer coverage. Further, tailoring products for every single user is not practical for a seller. For example, the sellers might have to shift entirely from what they were selling previously, just to fulfill a specific user's preference. Thus, it is not a cost-efficient solution from a supply chain perspective.

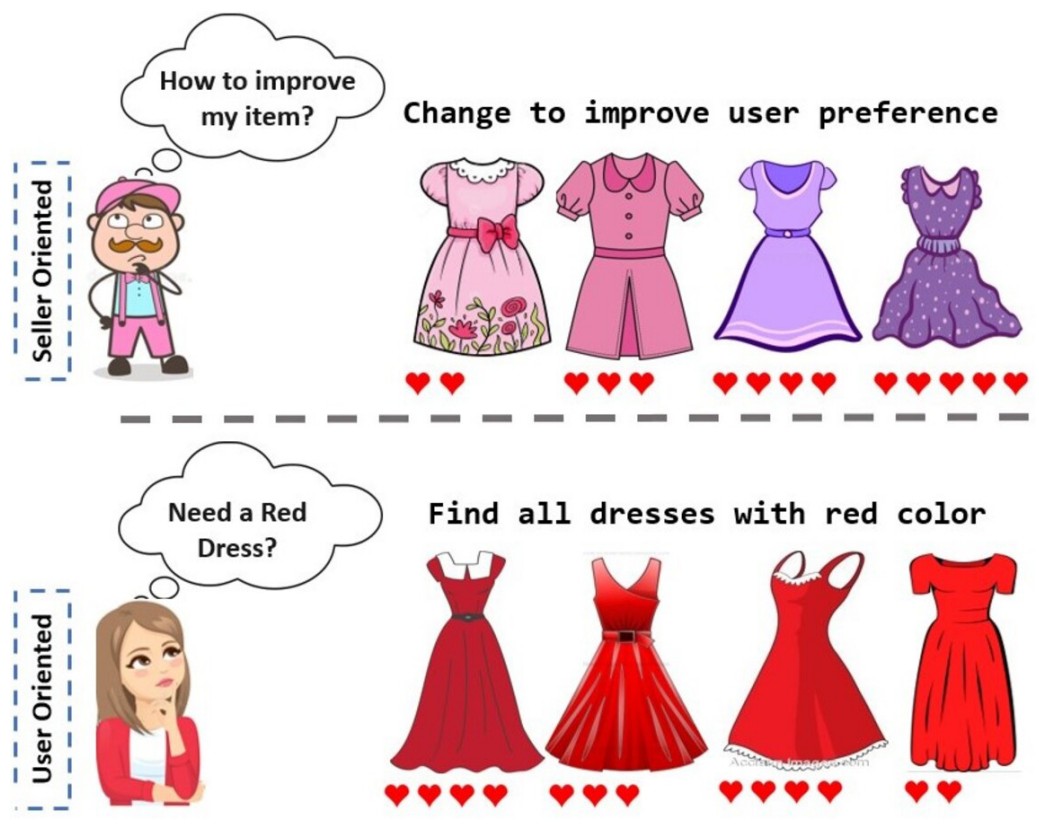

**Figure 1** **A seller-oriented approach provides feedback to sellers on how to modify their items by querying the recommendation model.** In contrast, a user-oriented approach presents users with their preferred items by leveraging the same model.

Also, it may not be feasible due to constraints on the seller's resources and machinery used to manufacture the products. Hence to solve such a problem, one should explore item-level solutions keeping the main item sold by the seller as the basis (Fig. 1). Such design modifications will lead to the generation of novel articles with similar physical dimensions as the base item but considerably distinct aesthetics, suiting the palette of a set of users.

Drawing inspiration from the work of *Kang et al. (2017)* and to bridge the gap between user preference and item-level feedback for sellers, we propose a novel framework to generate new product designs. Our method explores images in the vicinity of a specific item image embedded in a GAN's latent space to tackle the proposed problem of item-level image generation suited for a large set of users. Generating item images directly allows sellers to reference something when making modifications while also maintaining these changes without the ethical concerns associated with copying someone else's design. Therefore, our contributions are summarised as follows:

- We propose a framework to optimize distinct shifting vectors that are integrated with the base image's latent vector, enabling the generator to produce images that are physically closer in dimensions to the base item image, yet unique from each other.
- Our framework is generalized and can suggest item recommendations to a seller for any number of users, be it multiple or single.
- Our method is both generator and recommendation system agnostic, *i.e.,* it can work with any available pre-trained generators or recommender systems.
- To evaluate our method against the baseline, we use multiple metrics and conduct extensive experiments on the publicly available Amazon Fashion Dataset:

    – We compare the increase in preference scores achieved by our method and the baseline.
    – We analyze the rise in preference scores when both explored and unexplored items are used as anchor images for novel item generation across users.
    – Additionally, we present item-level modifications for various fashion object categories.

- Our strategy improves an item's user preference score by 16.7% when compared with the baseline.

## RELATED WORKS

The field of visually aware fashion recommendation involves modeling user behavior through signals captured from pictorial representations of user-interacted items. This field has been further segregated into two broad problem categories. First is the one that focuses on retrieving an actual fashion product by taking a user's preference history and the other relates to generating novel items using user-item relation.

To begin with the retrieval problem, *He & McAuley (2016)* were the first to introduce visual awareness in recommendation models by incorporating item images into the matrix-factorization model optimized using bayesian personalized ranking (BPR). To improve upon this, *Liu, Wu & Wang (2017)* argue that traditional visual recommendation systems require better modeling of an item's style, as they are often unable to distinguish between items having different styles but falling in the same category. To do this, the authors explicitly include item style derived from item & category latent representations in the BPR optimization setup to improve item recommendation to a user. Further, *Chen et al. (2017a)* with their method known as attentive collaborative filtering recall that the conventional collaborative filtering approach does not implicitly take into account whether the user is interested in the content on which he clicked. Thus, making the recommendation model give inaccurate recommendations. To solve this, authors employ an attention mechanism that weights different segments/regions of an item image or a video to determine how relevant the given item is to an user. Recently, *Hou et al. (2024)* investigated the zero-shot capabilities of large language models (LLMs) in item ranking for recommendation tasks, identifying key challenges such as their inability to grasp the chronological order of user-item interactions and a tendency to favor items listed earlier in the catalog while ranking. To address these issues, the authors proposed prompt engineering

techniques, demonstrating how LLMs, combined with their commonsense knowledge, can serve as an alternative to traditional recommendation models. In addition, a new sub-field has emerged in this category, *i.e.,* about retrieving compatible garments given an anchor garment. One such work, *Lin, Tran & Davis (2020)* proposes to use a nearest neighbor retrieval algorithm employed on an item's feature embedding to search for complimentary items that stylistically fit well with a given item. Another work *Lu et al. (2021)*, proposes self-attention to learn an outfit representation that captures high-order relationships between items that could fit together to make an outfit. This outfit representation is matched to a user style preference using a matching network to suggest compatible outfits. Shifting from previous works, *Sarkar et al. (2023)* proposed a unified framework that leverages self-attention to utilize the interactions among multiple outfit items to predict their overall compatibility and retrieve a missing item that aligns cohesively with both the existing outfit and the user-provided specification. Further, *Wang et al. (2023)* proposes a diffusion process that gradually corrupts and reconstructs user-item interactions by applying adjusted noise levels to predict interaction probabilities. Additionally, the authors incorporate latent and time-aware techniques to consider hidden factors while providing recommendations, such as item characteristics that affect user preferences and evolving dynamics of user behavior over time.

Shifting from these works which learn to retrieve items from existing catalog, *Kang et al. (2017)* for the first time introduced a generative setup using their method called deep visually-aware Bayesian personalized ranking (DVBPR). This work directly synthesized novel item images following a user's experience history and a class label. Additionally, *Phuoc Huynh et al. (2018)* presented another interesting problem of generating a complimentary product for a given item. Here, the authors propose to utilize a GAN setup that learns the co-occurrence of items to generate compatible item's latent representation which is then used to find a visually similar existing item using nearest neighbor search. Parallely, *Yu et al. (2019)* proposes to directly use a GAN setup for synthesizing complementary garment images, that fit well with a user's style preference and the given outfit.

Furthermore, there have been some other works that tackle slightly different problems such as transferring outfits to a given person's image, allowing users to directly manipulate physical attributes of an item, *etc. Zhu et al. (2017b)* is one such work where the authors propose a two-step image-segmentation based generative model that modifies an outfit by following an explicit textual description. *Han et al. (2018)* also proposes a GAN setup for solving a virtual try-on problem where an outfit is transferred to fit a user's body shape. Recently, *Baldrati et al. (2023)* proposed to utilize the diffusion process to assist users in generating fashion images by using garment sketches, textual descriptions or body pose images.

Though these advancements have enabled significant progress in the field of fashion recommendation/generation, a notable gap remains: they address the recommendation problem from a single perspective, predominantly the users. In contrast, our work aims to target the inverse by specifically focusing on how consumer behavior could complement a seller's product ideation step.

## BACKGROUND

### User preference

Visual recommendation systems utilize visual features extracted from an item image to model user preference. *Kang et al. (2017)* propose an end-to-end learnable (all parameters trained in one go) approach called deep-visually aware bayesian personalized ranking that learns dataset-specific visual features from scratch using a convolutional neural network (CNN) while focusing on an implicit feedback problem. In the context of implicit feedback, the only available information is whether a user has clicked on an item. Consequently, the recommendation problem reduces to ranking previously observed (clicked) items higher than unobserved (unclicked) items. The DVBPR (*Kang et al., 2017*) method formally describes the problem as follows: Given a dataset $D$ where $I$ denotes the set of items and $U$ denotes the set of users, we need to rank items set $I \setminus I_u^+$ with which a given user $u \in U$ has not interacted with. This ranking is done using the preference score $x_{u,i}$ which is the preference of user $u \in U$ towards item $i \in I_u^+$ using the item image $X_i$. This is mathematically represented by *Kang et al. (2017)* using the following equation:

$$x_{u,i} = \theta_u^T \phi(X_i), \tag{1}$$

Here $\theta$ represents the user embedding and $\phi$ represents the weights of a CNN architecture which are learned using the Bayesian personalized ranking (BPR) optimization framework (*Rendle et al., 2012*). BPR optimises the ranking using triplets $u, i, j \in D$, where:

$$D = \{(u, i, j) | u \in U \wedge i \in I_u^+ \wedge j \in I \setminus I_u^+\}, \tag{2}$$

Here $i \in I_u^+$ is an item observed by user $u \in U$ and $j \in I \setminus I_u^+$ is an unobserved item. Thus ideally, the preference score (Eq. (1)) of item $i$ should be more than item $j$, hence BPR objective function is defined as following:

$$\max \sum_{(u,i,j) \in D} \left( ln(\sigma(x_{uij})) - \lambda_\Theta ||\Theta||^2 \right), \tag{3}$$

Here $\sigma(\cdot)$ is the sigmoid function, $\lambda_\Theta ||\Theta||^2$ as the regularization term with $\lambda_\Theta$ as the regularization hyper-parameter. The $\Theta$ is the set of all the trainable weights of a neural network, and the term $x_{uij}$ is the preference score difference between item $i$ and item $j$ for a given user $u$ and is mathematically represented as follows:

$$x_{uij} = x_{ui} - x_{uj}. \tag{4}$$

### Perceptual similarity (LPIPS)

Given the seller-oriented focus of our work, we employ perceptual similarity as a key metric in our methodology. The learned perceptual image patch similarity (LPIPS) metric, as discussed by *Zhang et al. (2018)*, is foundational to our approach. LPIPS is based on the notion of human perceptual similarity, reflecting how humans assess the similarity between image patches. This metric is critical for evaluating and improving the alignment of fashion items with user preferences from a seller's perspective. Formally, the LPIPS

metric is defined as the $l_2$ distance between activations extracted from a stack of layers. The author uses activations ($w$) obtained from different layers ($l$) of CNN as it provides a better representational space for various tasks. LPIPS is mathematically defined as:

$$d(x, x_0) = \sum_l \frac{1}{H_l W_l} \sum_{h,w} ||w_l \odot (\hat{y}_{hw}^l - \hat{y}_{0hw}^l)||_2^2, \tag{5}$$

Here, for two image patches $x$ and $x_0$, we calculate the spatially averaged ($H * W$), $l_2$ distance aggregated over channel ($C$) between the $\hat{y}^l, \hat{y}_0^l \in R^{H_l * W_l * X_l}$. The $\hat{y}^l$ and $\hat{y}_0^l$ are obtained from unit normalized internal activations scaled (using Hadamard product [$\odot$]) channel-wise by $w_l \in R^{C_l}$ extracted from different layers ($l$) of a network $F$.

## METHODOLOGY

### Overview

Our method builds upon a pre-trained generative adversarial network (GAN) model (*Goodfellow et al., 2014*; *Karras et al., 2017*; *Brock, Donahue & Simonyan, 2018*). A GAN consists of two components: a generator and a discriminator. These models improve through adversarial training, where the generator aims to produce images that the discriminator cannot distinguish from real images, thereby maximizing the discriminator's loss. Conversely, the discriminator learns to accurately classify generated images as fake and real images as real, minimizing its classification loss. This adversarial process enables the generator to create images that do not exist in the original dataset, showcasing the GAN setup's ability to generate novel and realistic images.

This generative ability leads to their utilization in applications like image inpainting, facial attribute modification, and novel clothing generation. *Kang et al. (2017)* also utilizes the power of GANs to generate images for a given class of clothing to have a high preference score for a given user. This optimization process is mathematically expressed as:

$$\hat{\delta}(u, c) = \underset{e \in G(\cdot, c)}{argmax} \hat{x}_{u,e}, \tag{6}$$

$$\hat{x}_{u,e} = \theta_u^T \phi[G(z, c)] - \eta[D_c(G(z, c)) - 1]^2, \tag{7}$$

Here, Eq. (7) is the loss that contains the user preference score and discriminator loss of whether the generated image is real or fake. Thus, the author tries to optimize the generator such that it efficiently explores the latent data manifold for vectors corresponding to a given class leading to real and highly preferred item images. However, their work had a few limitations. First, on the methodology side, the actual generated images were of inferior quality. The reason for such low-quality images is that their problem formulation is much harder to solve because a GAN has to map a latent vector with both a given class and a given user to generate images. Also, the optimized objective is excessively biased toward maximizing user preference instead of image quality, which introduces noises that fool the recommendation models into believing that the image will be preferred by the user (*Brown et al., 2017a*; *Liu et al., 2019*). Moreover, at a given time the *Kang et al. (2017)* generator would only generate images that belong to a given class basis a user's preference. This

is hardly a situation in a real-life scenario, where a seller would want how he can make changes to his current products such that the modified product is ranked higher in the user preference list for a set of users (instead of a single user), doing so in a way that the modified image looks realistic.

These drawbacks motivated us to propose and build a framework that is more seller/manufacturer oriented allowing them to gain insights from their item consumer's preferences, and generalizable to any item by suggesting novel changes considering the item's current image. To do this, we utilize a pre-trained GAN architecture and explore concepts from both GAN inversion theory (*Abdal, Qin & Wonka, 2019*; *Abdal, Qin & Wonka, 2020*; *Bau et al., 2020*) and GAN latent disentanglement theory (*Voynov & Babenko, 2020*). GAN inversion deals with accurately back-mapping a given latent vector to an image, and GAN latent disentanglement refers to the exploration of useful directions within a GAN's latent space. One of the main reasons for us to use a pre-trained GAN architecture is to leverage its ability to generate high-quality/resolution images and solve our problem of generating realistic fashion images. Since we are using a GAN model that only takes in a latent vector as input, hence for a given item image, we need to find where this item is situated in the latent space of the GAN model which is very similar to the reverse diffusion. Thus, we build a process depicted in Fig. 2 where a latent vector is optimized to minimize a combination of perceptual loss defined in Eq. (5) and mean-square-error between the image generated from the optimized latent vector and the actual item image. The perceptual similarity loss helps in learning how two images are perceptually similar, and the mean-squared error loss helps in making the generated image look closer to the actual item image digitally.

After obtaining the latent vector, we fix its position and explore the surrounding latent space within a defined radius. This process involves optimizing a set of shifting vectors that when added to the base latent vector, produce images that are physically proximate to the base image yet exhibit aesthetic diversity. In addition to this, objective of this step is to maximize the user preference score while simultaneously minimizing diversity loss which indicates dissimilarity among shifting vectors, and shift prediction loss. To achieve this, our method utilizes a model composed of a ResNet18 feature extractor and several linear layers (as illustrated in Fig. 3) to calculate diversity loss and shift prediction. These metrics ensure the maintenance of uniqueness across the entire batch of shifting latent vectors.

By combining both components of base latent vector search and shifting vectors, our methodology aims to produce images closely resembling the given item image while catering to the preferences of user groups, thus enhancing its generalizability. Furthermore, our approach can also be customized for generating images at a global level by using randomly sampling base latent vectors, thereby bypassing the need for a seller image as the base. Similarly, by adjusting shifting vectors across a range of users rather than a single user, it becomes possible to identify items preferred by a broader user base without knowing about any item. These customizable enhancements improves the versatility and applicability of our methodology across diverse scenarios.

---

**Algorithm 1:** GIGS Training

**Initialise:** Generator $G(\cdot)$, Recommendation Feature Extractor $\phi(\cdot)$, Shift predictor $\Theta_s(\cdot)$, dataset $S$, training iterations $T$, batch size for shifting vector $n$, Targets ($Y \in \{0,\dots,n-1\}$), base latent vector learning rate $\alpha_1$, shifting vector learning rate $\alpha_2$, search radius $\epsilon$;

**for** *each data point* $\{U_i, X_i\} \in S$ **do**

    **Sample:** Base latent vector $z \in N(0,1)$;

    **for** *iterations = 1 to T* **do**

        $\hat{X}_z \leftarrow G(z)$; // Base Image
        $L_{lpips} \leftarrow LPIPS(X_i, \hat{X}_z)$;
        $L_{mse} \leftarrow (X_i - \hat{X}_z)^2)$;
        $L_{inv} \leftarrow L_{lpips} + L_{mse}$; // Inversion Loss
        $z \leftarrow z - \alpha_1 \cdot \nabla_z L_{inv}$; // Backpropagation
        $z \leftarrow \frac{z - \mu_z}{\sigma z}$; // Normalization

    **Sample:** Shifting vector batch $\{k\}^n \in N(0,1)$;

    **for** *iterations = 1 to T* **do**

        $k \leftarrow min(max(k, -\epsilon), \epsilon)$; // Clipping Shifting Vectors
        $k_z \leftarrow z + k$; // Shifting Base Latent Vector
        $k_z \leftarrow \frac{k_z - \mu_{k_z}}{\sigma k_z}$; // Normalization
        $\hat{X}_k \leftarrow G(k_z)$; // Generating Shifted Images
        $L_{pref} \leftarrow \theta_{U_i}^T \phi[X_k]$; // Preference Score
        $P_{shift}, P_{class} \leftarrow \Theta_s(X_k)$;
        $L_{class} \leftarrow L_{ce}(P_{class}, Y)$;
        $L_{shift} \leftarrow L_{mae}(P_{shift}, k)$;
        $L_s \leftarrow L_{class} + L_{shift} - L_{pref}$; // Shifting Loss
        $k \leftarrow k - \alpha_2 \cdot \frac{1}{n}\sum(\nabla_k L_s)$; // Backpropagation
        $\Theta_s \leftarrow \Theta_s - \alpha_3 \cdot \frac{1}{n}\sum(\nabla_{\Theta_s} L_s)$; // Backpropagation

    **Obtain:** $\hat{X}_k$ // Shifted Images

---

## Algorithm

To train our proposed methodology we use Algorithm 1 where we begin by initializing the pre-trained generator network $G(\cdot)$, Feature extractor $\phi(\cdot)$ obtained from recommendation model, Shift predictor $\Theta_s(\cdot)$, dataset $S$, no. of iterations $T$ to run the training process, no. of output images $n$, Targets $Y \in \{0,\dots,n-1\}$, learning rate of base latent vector $\alpha_1$, learning rate of shifting vector $\alpha_2$, and learning rate of Shift predictor $\alpha_3$

We then run our method for each randomly sampled user $U_i$ and a given item image $X_i$. We first begin by running the GAN inversion process where we first initialize base latent vector $z$ with normal distribution $N(0,1)$. Then for each iteration in range 1 to $T$, we first pass $z$ to the generator $G(\cdot)$ and obtain the image $\hat{X}_z$ corresponding to $z$. Afterward, we calculate perceptual similarity ($L_{lpips}$) and mean-squared error loss ($L_{mse}$) and optimize $z$ to minimize the summation of both of these losses. After the optimization, the optimized

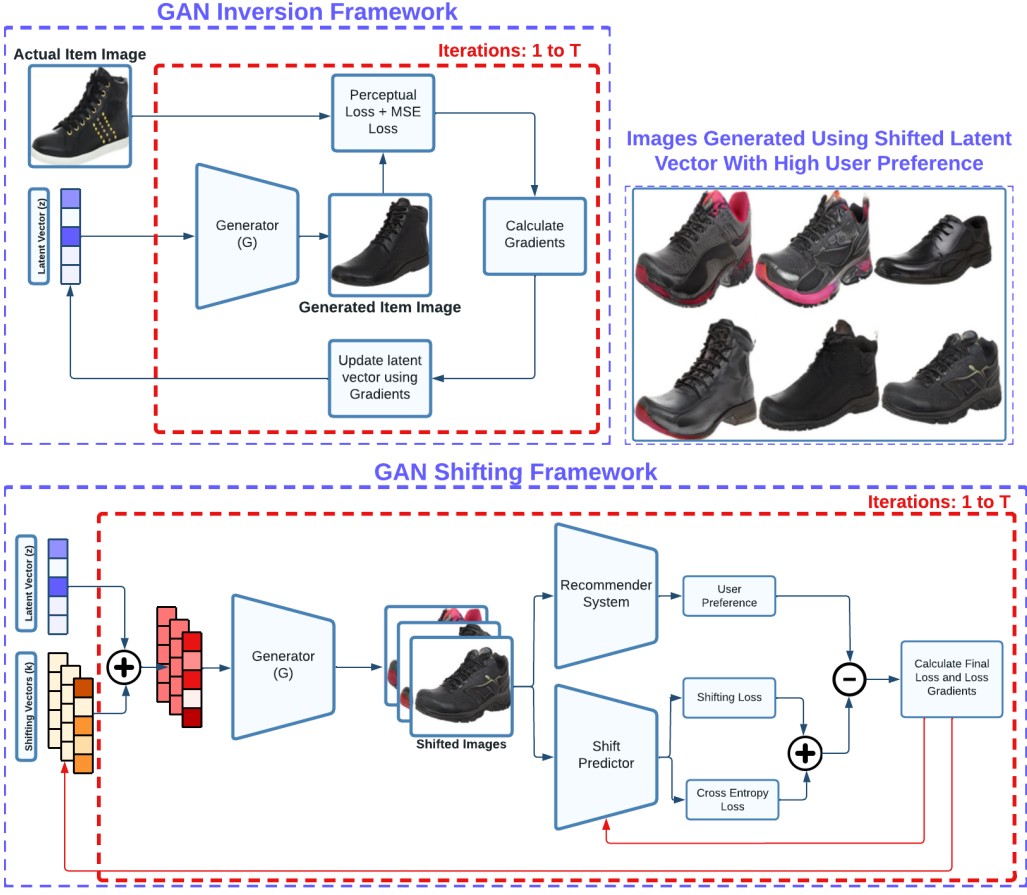

**Figure 2** **Our framework involves two main segments for generating item-level feedback for a vendor: GAN inversion and GAN shifting.** The GAN inversion segment optimizes a base latent vector, ensuring that the corresponding image generated by a GAN generator closely resembles the actual item image. The GAN shifting segment optimizes shifting vectors utilized to adjust the base latent vector, generating aesthetically distinct images that receive high preference across a set of users through cross-entropy loss minimization and preference score maximization.

$z$ may no longer be within the normal distribution. Hence we normalize it to have a mean of 0 and a standard deviation of 1.

After the first loop is finished, we obtain our final base latent vector $z$. We build upon $z$ to get a batch of shifting vectors $\{k\}^n$, when added to $z$ results, in images with high user preference. To do this we begin by initialising $\{k\}^n$ with normal distribution $N(0, 1)$. Then for each iteration in the range of 1 to $T$, we clamp the shifting vector to between a search radius of $\epsilon$. This clamping helps our process to only search for images that are somewhat similar to the image obtained using $z$. After this, we shift our base latent vector $z$ with batch $k$ to obtain a batch of shifted latent vectors $\{k_z\}$. Now we normalize the shifted latent vectors to have a normal distribution. This normalized variant of the shifted latent vector is then passed down to the generator $G(\cdot)$ to obtain shifted images $\hat{X}_k$, which we use to calculate the preference score $L_{pref}$ for the user $U_i$. To keep all the vectors present in shifting

**Figure 3** **Our shift predictor architecture is designed to predict both the shifted latent vector $P_{\text{shift}}$ and the corresponding class label $P_{\text{class}}$.** These predictions are used downstream to promote distinctiveness and separation among shifted latent vectors.

vector batch $\{k\}^n$ unique, we utilize a shift predictor $\Theta_s(\cdot)$ that learns to predict the $\{k\}^n$ as $P_{\text{shift}}$ and the probability $P_{\text{class}}$ with which each image in the batch $X_k$ lies in category $Y \in \{0, \ldots, n-1\}$. This category $Y$ is obtained using the batch size of shifting vector $\{k\}^n$. These values are then used to calculate the cross entropy loss $L_{ce}$ and mean absolute error $L_{mae}$. Finally, the batch of shifting vector $\{k\}^n$ and shift predictor $\Theta_s(\cdot)$ is optimized to minimize $L_s$, which is the summation of $L_{ce}$ and $L_{mae}$ from which we subtract $L_{pref}$. After

**Table 1  Amazon fashion dataset details.**

| Users | Items | Interactions | Categories |
|---|---|---|---|
| 64,583 | 234,892 | 513,367 | 6 |

completing all the iterations, we obtain the new shifted images ($\hat{X}_k$) that are closer to the base image and have high user preference.

## EXPERIMENTATION

### Dataset

We train, test, and compare our methodology on the Amazon fashion dataset (*Kang et al., 2017*; *McAuley et al., 2015*), which is a comprehensive real-world dataset with a large number of user-item interactions. Moreover, this dataset contains a large variety of item images mapped to 6 different classes (such as shirts, jackets, trousers, shoes, sandals, *etc.*) for both genders (male/female). Further details of the dataset are listed in Table 1.

### Training setup

Our methodology is designed to be versatile, *i.e.,* compatible with various GAN architectures applicable to real-world datasets, as exemplified by architectures such as ProGAN (*Karras et al., 2017*), Large Scale GAN (*Brock, Donahue & Simonyan, 2018*), and StyleGAN (*Karras, Laine & Aila, 2019*). However, through experimentation with the Amazon fashion dataset we observed that the ProGAN architecture proposed by *Karras et al. (2017)* consistently outperformed others in generating high-quality images. Although we also trained the StyleGAN architecture for the same dataset, the results were not comparable to those achieved with ProGAN. This discrepancy can be attributed primarily to the considerable variance and diversity of images within each class present in the Amazon fashion dataset. Note: All GAN architectures are trained using the default hyperparameters recommended by their respective authors.

For optimizing our base latent vector, shifting vector, and the shift predictor $\Theta_s(\cdot)$, we use the Adam optimizer with a learning rate of 0.01. Since the pre-trained ProGAN architecture takes the input of 512 dimensions, thus the size of both our base latent vector $z$ and shifting vector $k$ is set to 512. Additionally, each pixel in the output image generated by the pre-trained ProGAN model lies within the $[0, 1]$ range. Therefore, throughout our method, we maintain all image data within this $[0, 1]$ range. Further, to determine the optimal value for the search radius $\epsilon$, we start with 0.1 and increment it by 0.05. After each increment, we manually review the generated image results. We found that $\epsilon=0.4$ yields the best results, as values above this tend to generate images that deviate from the original class (*e.g.*, given a shoe image, the shifted image becomes a shirt). We also fixed batch size $n$ to 8 (can have any integer value) for all our experiments to reduce the time quantum to 1–2 min. We retrain the DVBPR recommendation model proposed by *Kang et al. (2017)*. To train, we use a single Nvidia RTX A6000. Finally, to compare our work, we use personalized and non-personalized variants of image generation methods proposed by *Kang et al. (2017)* as our baseline because it is the only work in the literature that has some

architectural similarities to our work, *i.e.,* directly generating an entire image given a user's preference. Both our method and *Kang et al. (2017)* method work at an image resolution of $(64 \times 64)$ scaled in later stages to $(128 \times 128)$ using bilinear interpolation.

## Evaluation

In this subsection, we discuss the various evaluation metrics used to evaluate the generated images. Through this evaluation, we want to focus on three main criteria which are: uniqueness, quality, and user preference.

### Inception score

In the GAN literature, this score (*Salimans et al., 2016*) is predominantly used to assess the quality of the generated images. It is based upon the output probability score provided by an inceptionV3 (*Szegedy et al., 2015*) ImageNet classifier. It is higher when the entropy/randomness is low in the output class probabilities, which means that the classifier can classify the given image into a single class. In other words, the generated image contains a single item or object, which makes the inceptionV3 classifier map it to only one of the single output imagenet classes.

### Opposite structural similarity index

Structural similarity index (SSIM) (*Wang et al., 2004*) score is used to evaluate the similarity in generated images. Similar to *Kang et al. (2017)*, we use it because this metric is more consistent with human perception than mean-square error. Since SSIM is between 0 to 1. By subtracting the SSIM score from 1 we obtain the opposite SSIM, which signifies the diversity in the generated images.

### Frechet inception distance

This score (*Heusel et al., 2017*) is a direct development over inception score. As explained above, the inception score only takes into account the generated images. However, the Frechet inception distance (FID) score considers both generated and ground truth images and matches stats (mean, std) obtained from the output distribution of the inceptionV3 classifier to determine the quality of the generated images. Since it uses the similarity between distribution stats as a quality measure, a low-value score is preferred.

### User preference

To quantify whether a user would prefer a generated image, we use the preference score represented by Eq. (1). A higher value of this measure would mean that the item would be ranked higher in the user preference list.

## Results

In this subsection, we first discuss the results of quantitative experiments done to verify the performance of our method when compared to DVBPR (*Kang et al., 2017*). We use the metrics described in 'Evaluation' to test the quality, diversity, and user preference of the generated images. To begin with, since *Kang et al. (2017)* is a more class-centric approach, *i.e.,* generates images that belong to a single class domain and are not item-specific, so we similarly change our methodology to have a suitable comparison. We remove our

**Table 2 Comparison of top-3 returned images for a given user (1,000 trials).** For all the metrics, a larger value is better except for FID. The value after the ± is the standard deviation of the score.

| Methodology | Preference score | Opposite SSIM | IS | FID |
|---|---|---|---|---|
| **Ours** | 8.97 ± 3.55 | 0.63 ± 0.16 | 8.22 ± 0.38 | 13.1 |
| DVBPR (Personalized) | 7.68 ± 4.1 | 0.53 ± 0.12 | 7.65 ± 0.24 | – |
| DVBPR (Non-personalized) | −1.94 ± 3.7 | 0.56 ± 0.09 | 6.81 ± 0.37 | - |

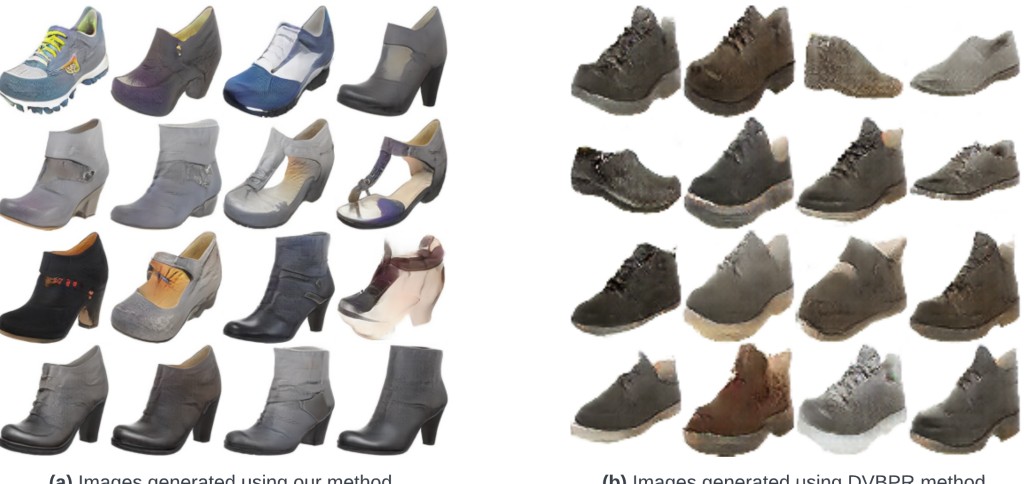

**(a)** Images generated using our method.     **(b)** Images generated using DVBPR method.

**Figure 4 Image quality comparison between our method and *Kang et al. (2017)*'s personalized method.**

item-specific portion from the framework defined in Fig. 2 and generate images around randomly sampled latent vectors using the ProGAN generator (*Karras et al., 2017*). After making both approaches comparable, we calculate the specified metrics, whose values are listed in Table 2.

Table 2 compares our method with two variants of DVBPR method (*Kang et al., 2017*). The first variant of the baseline is the personalized one, where the generator learns to optimize the user preference score, and the second one is the random one, where the main goal of the generator is to only optimize the quality of the generated images. As can be seen, our method obtains a mean preference score of 8.97, thus improving by 16.7% over the mean preference score of 7.68 obtained by the personalized variant of *Kang et al. (2017)*. Similarly, our methodology which is generating personalized images obtains a mean diversity (opposite SSIM) score of 0.63, thus improving by 12.5% over the non-personalized variant of *Kang et al. (2017)* and by 18.8% over the personalized variant of *Kang et al. (2017)*. Furthermore, our method outperforms (*Kang et al., 2017*) personalized method by 7.4%. Figure 4 compares the images generated for the above-explained experiment using our method and *Kang et al. (2017)* personalized method. Here, we see that, due to our proposed latent shifting: images generated using our method have a lot of variety in aesthetical aspects when compared to *Kang et al. (2017)*.

**Table 3** **Results of top-3 returned images for 1,000 randomly sampled users (with 1 randomly sampled preferred and unpreferred item).** For all the metrics, a larger value is better except for FID. The value after the ± is the standard deviation of the score.

| Item type | Preference score | Opposite SSIM | IS | FID | Preference shift |
|---|---|---|---|---|---|
| Un-preferred items | 5.48 ± 4.02 | 0.62 ± 0.16 | 8.51 ± 0.31 | 17.8 | 7.31 ± 3.04 |
| Preferred items | 7.40 ± 4.35 | 0.60 ± 0.16 | 8.25 ± 0.29 | 15.58 | 5.64 ± 2.63 |

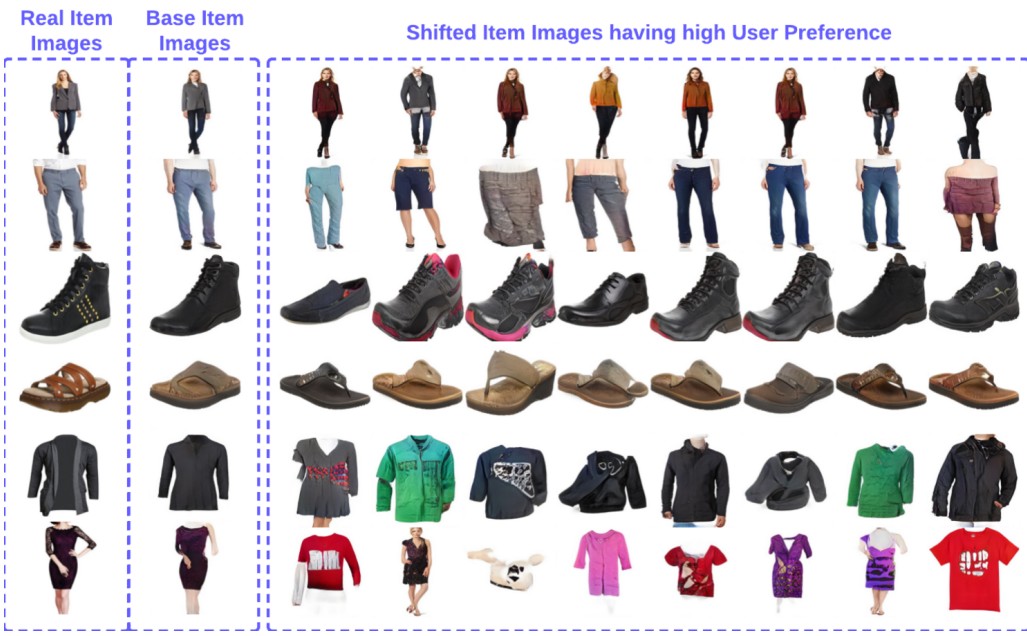

**Figure 5** **Item-specific image generation.**

Since our method can also work on item level, *i.e.,* has the ability to generate various images with high user preference for a randomly sampled user, we also perform item level experiments. Table 3 depicts the results of the item-level experiments, we perform with our method, where we first sample a randomly chosen user along with one preferred and one unpreferred item image and then generate novel images with our method. As can be seen in Table 3 for unpreferred items our method can generate unique, realistic images that obtain a mean incremental preference score shift (base image preference - generated image preference score) of 7.31. This signifies that in an ideal situation, any seller can use our method to make modifications to their item leading to improved rank in the user preference list. Qualitatively, Fig. 5 depicts the images generated around a given object with high user preference, as can be seen, our base item image is really close to the original item image. Moreover, there is a lot of variety in the shifted item image set aesthetically. Interestingly we also see a few images that are not related to the actual ground truth image. We believe that is happening because there is no constraint except for the search radius limitation on shifted images. Thus, a potential area to explore in the future.

**Table 4  Results of returned images for different subsets obtained from *McAuley et al. (2015)*.** For all the metrics, a larger value is better except for FID. The value after the ± is the standard deviation of the score. Across all the object categories, a consistent preference score shift of 7.3 is observed

| Subset type | Preference score | Opposite SSIM | IS | FID | Preference shift |
|---|---|---|---|---|---|
| Shirts | 5.08 ± 4.17 | 0.68 ± 0.15 | 7.24 ± 0.39 | 18.85 | 7.26 ± 3.11 |
| Footwear | 5.83 ± 4.03 | 0.63 ± 0.15 | 8.14 ± 0.34 | 16.91 | 7.54 ± 3.10 |
| Trousers | 5.48 ± 4.30 | 0.63 ± 0.18 | 8.24 ± 0.40 | 17.39 | 7.33 ± 3.08 |

**Table 5  Results of top-3 returned images for n = 1, 5, 10, 50, 100, 500, 1,000, 10,000 randomly sampled users.** For all the metrics, a larger value is better except for FID. The value after the ± is the standard deviation of the score.

| User count | Preference score | Opposite SSIM | IS | FID | Preference shift |
|---|---|---|---|---|---|
| $n = 1$ | 8.97 ± 3.55 | 0.63 ± 0.16 | 8.22 ± 0.38 | 13.10 | 7.31 ± 3.04 |
| $n = 5$ | 2.31 ± 1.51 | 0.61 ± 0.15 | 8.13 ± 0.35 | 17.65 | 4.37 ± 1.48 |
| $n = 10$ | 1.84 ± 1.18 | 0.61 ± 0.15 | 7.99 ± 0.36 | 17.95 | 3.77 ± 1.19 |
| $n = 50$ | 1.55 ± 0.88 | 0.60 ± 0.16 | 8.06 ± 0.26 | 17.45 | 3.06 ± 0.90 |
| $n = 100$ | 1.56 ± 0.89 | 0.60 ± 0.15 | 8.06 ± 0.25 | 17.66 | 3.22 ± 0.94 |
| $n = 500$ | 1.22 ± 0.84 | 0.60 ± 0.15 | 7.98 ± 0.23 | 17.56 | 3.12 ± 0.92 |
| $n = 1,000$ | 1.18 ± 0.85 | 0.60 ± 0.15 | 8.02 ± 0.22 | 17.43 | 3.15 ± 0.92 |
| $n = 10,000$ | 1.04 ± 0.82 | 0.60 ± 0.15 | 8.03 ± 0.43 | 17.12 | 3.12 ± 0.91 |

Additionally, we also perform a more real-life experiment to bring out the pragmatic capabilities of our approach. Firstly, we perform various tests on our approach against mutually exclusive datasets. For this, we divide the Amazon-fashion dataset (*McAuley et al., 2015*) into three major segments, *i.e.,* shirts, footwear, and trousers. Table 4 provides the results in line with metrics discussed in 'Evaluation'. As seen in Table 4 our approach provides a mean incremental preference score shift (base image preference–generated image preference score) of 7.3 across different subsets of data. Thus establishing that our approach is highly transferable.

Another major real-life scenario is where a seller has to modify its product such that the product has a high preference across a large number of users (not just one). To test our approach for such a scenario we randomly sample different subsets of users of size $n = \{1, 5, 10, 50, 100, 500, 1000, 10000\}$. Table 5 depicts the results of this experiment, Here one can see that our approach for user counts greater than 10 (Fig. 6) provides a constant mean incremental preference score shift (base image preference–generated image preference score) of 3.2 without affecting the quality of the image depicted by the FID, Inception Score metric.

## CONCLUSION & FUTURE WORK

We present a novel and versatile approach that leverages visual patterns learned by a recommendation system to discern consumer preferences. This information is then used to query a pre-trained generative model, producing novel images that push creative boundaries and serve as valuable feedback for item sellers. Extensive experiments demonstrate the high

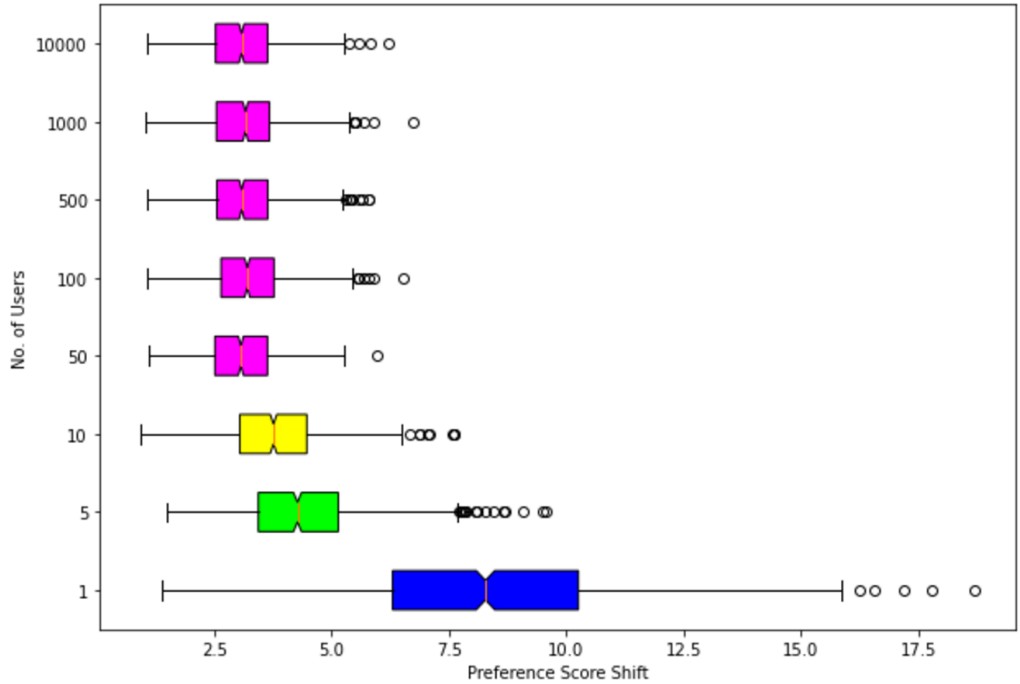

**Figure 6** **Box-plot for preference score shift across various sets of randomly sampled users n = 1, 5, 10, 50, 100, 500, 1,000, 10,000.** Similar color plots depict consistency in preference score shift.

generalizability of our method across diverse item categories, gathering preference insights from multiple users to generate naturalistic images centered around a single item. These images represent item-level enhancements, empowering sellers to expand their customer base. Moreover, our approach is model-agnostic, overcoming the limitations of specific generative or recommendation models, thus extending its utility beyond fashion or vision-based domains. In addition, this work marks an initial step toward seller-oriented feedback methodologies, offering numerous avenues for future refinement. For instance, latent diffusion models could be employed to enable controllable modifications, providing insights into how changes in an item are influenced by preferences for other product categories. Additionally, refining the GAN inversion process could improve the accuracy of generated outputs by constraining them to closely resemble the original item, effectively filtering out irrelevant object category associations. Looking ahead, incorporating privacy-preserving techniques such as differential privacy or selectively masking shifted latent codes could further enhance the method by safeguarding proprietary patterns while preserving critical insights for generating creative recommendations. These enhancements would ensure that our approach aligns with ethical standards and mitigates potential real-world concerns regarding intellectual property and applicability.

### Funding

The authors received no funding for this work.

### Competing Interests

The authors declare there are no competing interests.

### Author Contributions

- Satyadwyoom Kumar conceived and designed the experiments, performed the experiments, analyzed the data, performed the computation work, prepared figures and/or tables, authored or reviewed drafts of the article, and approved the final draft.
- Abhijith Sharma conceived and designed the experiments, performed the experiments, analyzed the data, performed the computation work, prepared figures and/or tables, authored or reviewed drafts of the article, and approved the final draft.
- Apurva Narayan conceived and designed the experiments, analyzed the data, authored or reviewed drafts of the article, and approved the final draft.

### Data Availability

The data and code is available at GitHub and Zenodo:

- https://github.com/Idsl-group/Gan-Inversion-and-Shifting-GIS

- SATYADWYOOM KUMAR. (2024). Idsl-group/Gan-Inversion-and-Shifting-GIS: GIS (GAN Inversion & Shifting) (v0.0.0). Zenodo. https://doi.org/10.5281/zenodo.13630810.

https://zenodo.org/records/14540482

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
