# Peer review of "GAN inversion and shifting: recommending product modifications to sellers for better user preference"

_PeerJ Computer Science, doi:10.7717/peerj-cs.2553_

## Round 0.1 · original submission · Major Revisions

Dear authors,

Thank you for submitting your article. Based on reviews' comments, your article has not yet been recommended for publication in its current form. However, we encourage you to address the concerns and criticisms of the reviewer and to resubmit your article once you have updated it accordingly. Before submitting the paper following should also be addressed:

1. The Abstract section should be rewritten in order to clearly state the manuscript's main focus. The abstract should give the readers essential details, i.e., including the main contributions, the proposed method, the main problem, the obtained results, the benchmark tests and data sets, the comparative methods, etc. Efforts are needed to make the abstract coherent while clearly describing the problem being investigated and findings.
2. The current Introduction is simple and misses many contents related to the problem formulation. There is not a clear categorization of related work.
3. Please expand “Conclusion” section with some specific ideas for future work.

Best wishes,

Reviewer 1 ·

Basic reporting

Overall, the paper is well organized and well presented.

The writing quality should be significantly improvement. Below are some examples of language misuse:
* "However" is used in multiple places where there is no contrast or dispute of a previous opinion.
* It is strange to see something like "authors in Kang et al. (2017) ..."

Experimental design

The paper explores the research topic of suggesting modification/refinement of products to sellers that could potentially better align with user preferences. The problem is quite interesting while it is slightly different from the problem in Kang et al. (2017).

The methodology introduced is kind of too brief. Especially, Figure 2 lacks lots of detailed explanations to help readers fully understand the framework.

Validity of the findings

The necessity of new image generation should be justified. Instead of generating images to give suggestions, is it possible or more convenient to directly give instructions to users regarding how to modify the existing items, or it is possible to simply retrieve an image from the existing item base to provide such suggestions?

Experiments were conducted on only one Amazon dataset. It is preferable that more datasets are used to validate the soundness/advantages of the proposed framework.

Besides, only one baseline method is used in the experimental comparison, making the results not so solid.

Reviewer 2 ·

Basic reporting

Different from conventional recommendation system, this paper purposed a new method to enhancing user experience by recommending item refinements to sellers. Specifically speaking, They generate novel images based on the item images hung on platforms with a GAN network and feedback to item sellers. It is indeed a novel approach, but there are still some weaknesses:
1. The method of comparison in the experiment is a paper purposed in 2017.
2. Consider that your approach is actually an image generator, can you explain the advantages of choosing GAN instead of the diffusion model which is more mainstream currently?
3. Please add references to related work
4. Formulas and symbols need to be corrected to conform to correct specifications.
5. Lack of the latest comparison methods, need to increase the relative comparison methods.

Experimental design

no comment

Validity of the findings

no comment

Additional comments

no comment

Reviewer 3 ·

Basic reporting

The paper is well-written and adheres to professional standards of English. The introduction effectively sets the context and motivation for the study, emphasizing the limitations of conventional recommendation systems and introducing the novel approach of providing feedback to sellers.

Suggested Improvements:

The literature review, while comprehensive, can benefit from including more recent works to provide a broader context.

Ensure that all terms and theorems introduced are clearly defined, and proofs are detailed where necessary to enhance understanding.

Experimental design

The experimental design aligns well with the aims and scope of the journal. The investigation is rigorous, with a high technical standard.

Suggested Improvements:

The methods section should provide more detailed information on data preprocessing steps to ensure reproducibility. Clarifying the specifics of how data was preprocessed, including any normalization or augmentation techniques used, would be beneficial.
The evaluation methods and metrics are well-chosen, but a more detailed explanation of the model selection methods and hyperparameter tuning would be helpful.
Ensure that all sources are adequately cited, with proper quotations or paraphrasing as appropriate.

Validity of the findings

The paper's conclusions are well-supported by the experimental results. The arguments are coherent and align with the goals outlined in the introduction.

Suggested Improvements:

The discussion section should explicitly address any limitations of the study and propose future directions for research. Identifying unresolved questions and potential areas for further investigation would provide a more comprehensive conclusion.
Ensure that the experiments and evaluations are thoroughly described, with sufficient detail for meaningful replication. Any assumptions made during the study should be clearly stated and justified.

Additional comments

Overall, the paper presents a significant contribution to the field of recommendation systems, particularly in providing actionable feedback to sellers based on user preferences. The methodology is innovative and well-executed, offering a practical solution to a real-world problem.

General Comments:

The paper would benefit from a stronger emphasis on the practical implications of the proposed method. Discussing how this approach can be integrated into existing e-commerce platforms and the potential impact on both sellers and users would add value.
Consider including more visual examples of the generated images and their corresponding user preference scores to illustrate the effectiveness of the proposed framework.
Thank you for your attention to these comments and suggestions. I believe addressing these points will significantly enhance the quality and impact of the paper.

---

## Round 0.2 · Minor Revisions

Dear authors,

Thank you for the revised paper. One of the previous reviewers did not respond to the invitation for revision. Generally, your paper seems to be sufficiently improved. However, please make the necessary changes and additions suggested by Reviewer2 and Reviewer3 and resubmit the paper.

Best wishes,

Reviewer 2 ·

Basic reporting

1、All formulas need punctuation after them.
2、Lack of discussion around related work.

Experimental design

1、The contributions in this article are too scattered, and it is suggested to summarize them into 3-4 points

Validity of the findings

no comment

Additional comments

no comment

Reviewer 3 ·

Basic reporting

1. Basic Reporting
The manuscript is well-written and clear. The authors provide a thorough background on recommendation systems, generative adversarial networks (GANs), and their application in product image modification. The literature review covers foundational works, such as the introduction of GANs by Goodfellow et al. (2014) and subsequent improvements by Karras et al. (2017), but it would benefit from additional recent references in the field of GAN-based item generation.

Suggested Improvements:

Incorporate more recent studies on recommendation systems and GAN-based item generation.

Figures and tables are presented clearly, but captions could be enhanced with more descriptive details.

Experimental design

2. Experimental Design
The experimental design is solid, with a detailed description of the GAN inversion and shifting method. The proposed methodology focuses on seller-oriented recommendations by suggesting product modifications based on user preferences. The paper effectively explains the GAN inversion process, the shifting of latent vectors, and the use of perceptual similarity (LPIPS) as a key metric.

Suggested Improvements:

Provide more details on the dataset preprocessing steps, such as how the data was normalized or cleaned for training.
Clarify the hyperparameter tuning for the GAN models and explain how the search radius for shifting vectors was selected.

Validity of the findings

3. Validity of the Findings
The findings are well-supported by the experimental results, demonstrating that the proposed GAN inversion and shifting method improves user preference scores by 16.7% compared to the baseline. The results are presented through quantitative measures like Inception Score (IS), SSIM, and FID, as well as qualitative comparisons between the generated images.

Suggested Improvements:

Discuss potential limitations of the approach, such as its generalizability to different product categories or its scalability for large datasets.
Include more details about future research directions, such as incorporating real-world seller feedback or extending the method to more complex item categories.

Additional comments

4. Additional Comments
The paper presents a valuable contribution to the field of GAN-based item generation and recommendation systems. By focusing on the seller’s perspective, the authors introduce a novel approach that bridges the gap between customer-centric recommendation systems and seller-oriented feedback.

General Comments:

Consider discussing practical implications of the proposed approach, such as its application in e-commerce platforms where sellers can modify product designs based on real-time user preferences.
A section on the ethical implications of using GANs in product design, including considerations about intellectual property and the potential for misuse, would add depth to the discussion.
Thank you for the opportunity to review this manuscript. The research is innovative, and the suggestions provided aim to improve its impact and clarity.

---

## Round 0.3 · accepted · Accept

Dear Authors,

Thank you for the revised paper. I am grateful to you for responding to the reviewers' comments in a comprehensive and satisfactory manner, and for making the requisite additions and modifications. I am now in a position to accept your paper.

Best wishes,

Reviewer 2 ·

Basic reporting

The manuscript solved my doubts and problems.

Experimental design

N/A

Validity of the findings

N/A

Additional comments

N/A

Reviewer 3 ·

Basic reporting

The manuscript is well-written, with clear, professional language. The authors introduce a novel approach to recommendation systems, aiming to provide item-level feedback to sellers based on user preferences. The literature review comprehensively covers key works in recommendation systems and generative adversarial networks (GANs), but it could benefit from more recent references, especially those relevant to GAN-based recommendation.

Suggested Improvements:

The literature review would be strengthened by including recent studies on GAN applications in recommendation systems and item generation.

Figures and tables are clear, but enhancing captions with additional detail could improve readability, particularly for those less familiar with GAN models and recommendation metrics.

Experimental design

The experimental design is robust and well-documented, using the Amazon Fashion dataset to validate the model’s ability to enhance user preference scores by generating item modifications. The authors effectively integrate GAN inversion techniques with shifting vectors, enabling the creation of images optimized for higher user preference.

Suggested Improvements:

Include more details on data preprocessing, such as handling missing values or outliers, to enhance reproducibility.
Provide additional clarity regarding hyperparameter tuning for both the GAN and shifting vector components, including parameters such as the radius of shifting vectors and their selection rationale.

Validity of the findings

The results are supported by extensive experiments, demonstrating that the proposed method outperforms baseline models, achieving a 16.7% increase in user preference score. The authors use a variety of evaluation metrics, including Inception Score (IS) and Frechet Inception Distance (FID), to validate the quality and diversity of generated images, which further supports the effectiveness of their model.

Suggested Improvements:

Discuss potential limitations of the approach, particularly in its scalability to larger datasets or different product categories.
Consider adding a section on potential future work, such as experimenting with other datasets or extending the methodology to accommodate non-fashion items.

Additional comments

This paper makes a valuable contribution to recommendation systems by shifting focus from a user-centric to a seller-oriented model. By providing sellers with actionable feedback on item modifications, the research offers practical insights with potential real-world applications in e-commerce.

General Comments:

Consider exploring the practical applications of the model in e-commerce platforms, such as how it could assist in product design or real-time modification feedback.
Including a discussion on the ethical considerations of using GANs for product modification, especially regarding intellectual property and design originality, would add depth to the paper.
Thank you for the opportunity to review this manuscript. The suggestions provided aim to enhance the quality and relevance of this innovative research.